# COVID-19 Vaccination in People Living with HIV (PLWH) in China: A Cross Sectional Study of Vaccine Hesitancy, Safety, and Immunogenicity

**DOI:** 10.3390/vaccines9121458

**Published:** 2021-12-09

**Authors:** Ying Liu, Junyan Han, Xin Li, Danying Chen, Xuesen Zhao, Yaruo Qiu, Leidan Zhang, Jing Xiao, Bei Li, Hongxin Zhao

**Affiliations:** 1Clinical Center for HIV/AIDS, Clinical and Research Center of Infectious Diseases, Beijing Ditan Hospital, Capital Medical University, Beijing 100015, China; liuying9509@mail.ccmu.edu.cn (Y.L.); libei@ccmu.edu.cn (B.L.); 2Beijing Key Laboratory of Emerging Infectious Diseases, Institute of Infectious Diseases, Beijing Ditan Hospital, Capital Medical University, Beijing 100015, China; hanjunyan@ccmu.edu.cn (J.H.); chendanying@ccmu.edu.cn (D.C.); zhaoxuesen@ccmu.edu.cn (X.Z.); 3Department of Center of Integrated Traditional Chinese and Western Medicine, Beijing Ditan Hospital, Capital Medical University, Beijing 100015, China; leaxin@ccmu.edu.cn; 4Clinical Center for HIV/AIDS, Clinical and Research Center of Infectious Diseases, Peking University Ditan Teaching Hospital, Beijing 100015, China; 2011210686@bjmu.edu.cn (Y.Q.); 2011210683@bjmu.edu.cn (L.Z.); 2011210598@bjmu.edu.cn (J.X.)

**Keywords:** HIV, COVID-19, SARS-CoV-2 vaccine, vaccine hesitancy, safety, immunogenicity

## Abstract

The administration of COVID-19 vaccines is the primary strategy used to prevent further infections by COVID-19, especially in people living with HIV (PLWH), who are at increased risk for severe symptoms and mortality. However, the vaccine hesitancy, safety, and immunogenicity of COVID-19 vaccines among PLWH have not been fully characterized. We estimated vaccine hesitancy and status of COVID-19 vaccination in Chinese PLWH, explored the safety and impact on antiviral therapy (ART) efficacy and compared the immunogenicity of an inactivated vaccine between PLWH and healthy controls (HC). In total, 27.5% (104/378) of PLWH hesitated to take the vaccine. The barriers included concerns about safety and efficacy, and physician counselling might help patients overcome this vaccine hesitancy. A COVID-19 vaccination did not cause severe side effects and had no negative impact on CD4^+^ T cell counts and HIV RNA viral load. Comparable spike receptor binding domain IgG titer were elicited in PLWH and HC after a second dose of the CoronaVac vaccine, but antibody responses were lower in poor immunological responders (CD4^+^ T cell counts < 350 cells/µL) compared with immunological responders (CD4^+^ T cell counts ≥ 350 cells/µL). These data showed that PLWH have comparable safety and immune response following inactivated COVID-19 vaccination compared with HC, but the poor immunological response in PLWH is associated with impaired humoral response.

## 1. Introduction

The rapid spread of coronavirus disease 2019 (COVID-19) caused by severe acute respiratory syndrome coronavirus 2 (SARS-CoV-2) led to significant morbidity and mortality as well as substantial psychological and economic costs worldwide [1]. The COVID-19 pandemic has led to decreased access to HIV-prevention services, HIV testing, HIV treatment and viral suppression, which could lead to less control over the HIV epidemic [2]. People living with HIV (PLWH) have been disproportionately affected by COVID-19 and are at increased risk for severe clinical symptoms and mortality due to SARS-CoV-2 infection, especially among those with lower CD4^+^ T cell counts or unsuppressed HIV viral replication [3,4,5,6].

The administration of a COVID-19 vaccine is considered the most effective and economic way to prevent infection by COVID-19 and to control its spread. Central to achieving high levels of vaccination coverage needed to effectively control the spread of COVID-19 is overcoming vaccine hesitancy [7]. However, attitudes toward COVID-19 vaccines and potential risk factors of vaccine hesitancy have not yet been well characterized.

Several studies have explored the reasons for COVID-19 vaccine hesitancy in a general population, with vaccine-specific concerns (side effects and efficacy) being the most commonly cited [8,9,10,11]. A French study showed that emphasizing the collective benefits of herd immunity and reassuring the safety of the proposed COVID-19 vaccine to PLWH is important to minimize vaccine hesitancy [12]. To date, two inactivated vaccines are widely used in China (CoronaVac vaccine and BBIBP-CorV vaccine), with satisfactory safety and immunogenicity among the general population in clinical trials [13,14]. Based on a low theoretical risk and the high potential benefit of vaccination, a panel convened by the Chinese Association of Infectious Diseases recommended that PLWH with suppressed viral load be immunized with a COVID-19 vaccine as soon as possible [15]. However, with limited information on vaccine safety and limited efficacy data available but noting their increased risk, PLWH may have conflicted COVID-19 vaccine attitudes.

To address this lacuna, we initiated a questionnaire-based survey to explore issues surrounding COVID-19 vaccine hesitancy in this vulnerable population. Additionally, we sought to explore the safety experiences, including the impact on the efficacy of ART among those who had already been vaccinated with the first dose, and to learn about the immunogenicity of the CoronaVac vaccine in PLWH and health controls (HC), as this might provide information useful for combating hesitancy.

## 2. Materials and Methods

This was a cross-sectional, observational study. The survey was conducted in an out-patient clinic of Beijing Ditan Hospital, Capital Medical University, a large hospital designated for treating the COVID-19 pandemic and HIV infections, to investigate vaccination statuses, willingness to be vaccinated, and adverse reactions towards COVID-19 vaccines. Most patients who visited the out-patient clinic were followed up every 6 months to perform CD4^+^ T cell counts and HIV RNA viral load (VL) testing and were prescribed ART. PLWH were eligible if they met the following inclusion criteria: (1) 18–60 years old; (2) have been receiving a stable ART regimen for at least 1 year with an VL ≤ 50 copies/mL; (3) have no COVID-19 infection history and no contact history, including close or indirect contact with a person with a confirmed COVID-19 infection; (4) completed the questionnaire; and (5) signed written informed consent.

We used two methods to recruit participants. We approached patients in the out-patient clinic in person and invited them to participate. If they agreed and were eligible, we provided a private room in which the participants and the research assistant could interact, and participants then completed a paper-based questionnaire. On the other hand, we recruited age- and sex-matched HC who had been vaccinated with two doses (0.5 mL/dose) of CoronaVac (Sinovac Life Sciences, Beijing, China) for at least 2 weeks by advertisements on the Internet. Plasma samples of the PLWH and HC were collected to measure humoral response to SARS-CoV-2 anti spike receptor binding domain-protein (S-RBD). The participants were recruited from 20 July to 4 August 2021 and the data were collected from 20 July to 20 August 2021. The study flow diagram is shown in Figure 1.

The study was approved by the Human Science Ethical Committee of Beijing Ditan Hospital, Capital Medical University (No. 2021-021-02). Participation was voluntary, and completion of the questionnaire implied consent for study participation. All information gathered was anonymized and kept confidential.

The questionnaire was completed by PLWH with assistance from the researcher. It involved three items: (1) demographics, HIV characteristics, and health status; (2) perception of COVID-19 vaccination; and (3) vaccination status and safety of the COVID-19 vaccine. The demographics included gender, age, marital status, educational background, and occupation. The HIV characteristics included duration of ART treatment, mode of HIV transmission, CD4^+^ T cell counts, and VL prior to ART initiation and 6 months ago. Health status was measured using 12-item short form health survey (SF-12), which is a 12-item questionnaire of which the answers allow for the calculation of Physical Component Summary (PCS) and Mental Component Summary (MCS) scores [16]. We assessed intent to be vaccinated for SARS-CoV-2 using the question, “Have you been vaccinated against COVID-19?”, followed by the response options “Yes” and “No”. Participants who responded “No” were asked the following multiple-choice question: “What is preventing you from becoming vaccinated?”. The response options were “Afraid of the side effects and/or poor efficacy”, “Contraindications for the vaccine”, “No perceived need for vaccination”, “Waiting to be scheduled”, and “Scheduling conflicts”. Participants who responded “Yes” were asked the following open-ended question: “Do you have any concerns after vaccination? If so, please specify.” To explore the role of physicians in encouraging vaccine acceptance, all participants were asked whether they discussed the COVID-19 vaccine with their physicians. PLWH who were vaccinated with at least one dose were asked to provide the details of vaccination, including the manufacturer of their COVID-19 vaccine, the date of each dose, and any adverse reactions that occurred within 28 days after each dose. The safety of the COVID-19 vaccine was assessed by local (pain, swelling, redness, and itching) and systemic (fever, fatigue, diarrhea, muscle pain, nausea, headache, vomiting, cough, joint pain, and hypersensitivity) adverse events. The adverse events reported were graded according to the China National Medical Products Administration guidelines [17]. The causal association between adverse events and vaccination was determined by the investigators. The survey items are shown in Appendix A.

We used ELISA kits (Wantai BioPharm, Beijing, China) to evaluate the spike receptor binding domain-protein specific IgG (S-RBD-IgG) antibody titers according to the manufacturer’s protocol. Briefly, the plasma was inactivated at 56 °C for 30 min to safety considerations, diluted 11-fold, and then applied to 96-well plates coated with purified SARS-CoV-2 RBD protein for 30 min at 37 °C. After washing five times, antibody binding was revealed using anti-human IgG labeled by HRP. Subsequently, the substrate solution and stop solution were added sequentially, and the plate absorbance was read at 450 nm and 630 nm after the reaction stopped. The optical density (OD) values were then converted into the equivalent enzyme units (U/mL) using a standard curve derived from known concentrations of a SARS-CoV-2-IgG antibody standard.

The maximum percentage of missing values did not exceeded 5% (3.2%, *n* = 12) in the present study and the missing values were excluded from analysis [18]. The characteristics of the survey respondents were summarized using frequencies (percentages) or medians (interquartile intervals, IQR). We used crosstabulations and chi-square tests to estimate the unadjusted associations between participant characteristics with the intent to become vaccinated. The Mann–Whitney U test was performed to compare continuous variables, and the chi-square test was chosen to test associations between vaccine willingness and categorical predictor variables. Paired continuous variables were compared using the Wilcoxon signed rank test. A *p*-value of 0.05 or lower was considered statistically significant. Statistical analysis was conducted using SPSS, version 26.0 (IBM Corp, Armonk, NY, USA), GraphPad Prism, version 8.0.1 (GraphPad Software, La Jolla, CA, USA) and R studio, version 4.0.3 (R Core Team, Vienna, Austria, 2020).

## 3. Results

### 3.1. Demographics, HIV Characteristics, and Health Status of PLWH

A total of 383 PLWH were recruited, and 378 questionnaires were available (response rate was 98.7%). The participants consisted of 374 males (98.9%) and 4 females (1.1%), with a median age of 34 years (IQR 30–39, Table 1). The majority of participants (51.3%) were 31–40 years old. They had varied levels of educational attainment, with more than two thirds (67.2%) having a college or undergraduate diploma. Most participants were unmarried (70.0%). Business/service staff (35.1%) accounted for the highest occupational group, followed by professional and technical personnel (20.7%), public officials (9.8%), and farmers/workers (5.7%).

All PLWH had undetectable plasma VL (<50 copies/mL) for at least 6 months and had received efavirenz (EFV), tenofovir disoproxil fumarate (TDF), and lamivudine (3TC) for at least 1 year without interruption. The median CD4^+^ T cell counts and VL prior to ART initiation was 305 cells/µL (IQR 203–433) and 4.67 log_10_ copies/mL (IQR 4.16–5.01), respectively. Their median duration of ART treatment was 4.3 years (IQR 2.8–6.0), and the median CD4^+^ T cell counts 6 months ago was 578 cells/µL (IQR 428–725). Men who have sex with men (MSM, 72.7%) were the major HIV transmission risk group in this cohort, followed by those with other or unknown transmissions (16.1%). Median scores of PCS and MCS in PLWH were 53 (IQR, 47–55) and 53 (IQR, 46–56), respectively. Detailed results are presented in Table 1.

### 3.2. COVID-19 Vaccination Status, Intention to Receive COVID-19 Vaccine, and Potential Risk Factors of Vaccine Hesitancy in PLWH

Among the 378 PLWH, 219 (57.5%) PLWH received at least one dose of a vaccine and 159 (42.5%) PLWH have not been vaccinated. In the vaccinated population, 70.7% (155/219) completed whole-course vaccination, of which nearly all (94.9%) were vaccinated with an inactivated vaccine (CoronaVac vaccine: 58.4% and BBIBP-CorV vaccine: 39.7%). Only 16 vaccinated PLWH had concerns about safety and/or the efficacy of the vaccine, and one was afraid of information leakage regarding HIV infection (Table 2). Next, we explored the reasons for not becoming vaccinated among the 157 unvaccinated patients (Table 3). Concerns about the side effects and/or poor efficacy of the vaccine was the most common reason (56.0%), followed by waiting to be scheduled (19.5%), having contraindications for the vaccine (13.8%), no perceived need for vaccination (9.4%), and scheduling conflicts (1.9%).

Patients who hesitated to become vaccinated are defined as those worried about safety and/or efficacy and described vaccination as being unnecessary. Overall, 27.5% (104/378) of PLWH hesitated to receive a COVID-19 vaccine. The univariate analysis shows that age, marital status, educational background, occupation, duration of ART treatment, CD4^+^ T cell counts before 6 months, CD4^+^ T cell counts and VL prior to ART initiation, and SF-12 scores had no impact on vaccination willingness (all *p* > 0.05, Table 4), while the number of PLWH who consulted their physicians about the COVID-19 vaccine was significantly lower in those with vaccine hesitancy (36.5% vs. 54.0%, *p* = 0.002). These results demonstrate that concerns about the safety and efficacy of the vaccine are major obstacles to COVID-19 vaccination. Physicians play an important role in encouraging vaccine acceptance among PLWH.

### 3.3. Safety and the Impact on ART Efficacy of the COVID-19 Vaccine on ART Efficacy in PLWH

In total, 215 PLWH have received at least one dose of an inactivated vaccine. Approximately one third (33.4%) reported at least one adverse reaction within 28 days after each dose. For the CoronaVac vaccine and the BBIBP-CorV vaccine, the incidence of adverse reactions was 35.1% and 31.2%, respectively. The most common local adverse reaction was injection site pain (25.1%, Figure 2A). The most common systemic adverse reaction was fatigue (13.5%), followed by fever (4.7%) and headache (3.3%). All adverse reactions were mild (grade 1 or grade 2) and self-limited. These results suggest that the COVID-19 vaccine had a good safety profile.

In order to explore the impact of a COVID-19 vaccine on ART efficacy in PLWH, we compared the CD4^+^ T cell counts and VL 6 months ago versus the results during this visit. The median CD4^+^ T cell counts of vaccinated patients was 580 (447–723) cells/µL before vaccination and significantly increased to 604 (452–752) cells/mL after vaccination (*p* = 0.035, Figure 2B), while the CD4^+^ T cell counts of unvaccinated people did not change markedly (578 (420–758) cells/µL vs. 562 (420–734) cells/µL, *p* = 0.752). No event of viral rebound (>50 copies/mL) was reported. Since residual viremia below 50 copies/mL has been associated with a higher risk of virologic failure in previous studies [19], we further confirmed that no significant difference was found in the proportion of VL remaining to be “target not detected” (TND) between the vaccinated and unvaccinated group (91.9% vs. 94.3%, *p* = 0.412, Figure 2C), suggesting no negative impact of COVID-19 vaccine on ART efficacy.

### 3.4. The CoronaVac Vaccine Elicited Comparable Antibody Responses in PLWH Compared with HC

To investigate the humoral responses to COVID-19 vaccines, we recruited 55 PLWH and 21 age- and sex-matched HC who completed vaccination with two doses of the CoronaVac vaccine for at least 2 weeks (ranging from 2 to 18 weeks) and measured their plasma S-RBD-IgG antibody titers. As expected, PLWH had lower CD4^+^ T cell counts compared with HC (572 ± 203 cells/µL vs. 769 ± 262 cells/µL, *p* = 0.001, Table 5). All PLWH had a CD4^+^ T cell count of above 200 cells/µL before vaccination. The median time interval between administration of the second dose and blood collection and the vaccination interval between two doses were comparable between groups (*p* = 0.921; *p* = 0.969, respectively). After the whole-course vaccination, the S-RBD-IgG titers were similar in PLWH and HC (15.8 U/mL (IQR,10.4–23.3) vs. 16 U/mL (IQR, 11.3–23.2), *p* = 0.839, Figure 2D), and the two groups had a similar dynamic curve for S-RBD-IgG titers (Figure 2E). Therefore, a similar immunogenicity of CoronaVac was noted in PLWH compared with HC.

### 3.5. Poor Immunological Response Was Associated with Impaired Antibody Responses to CoronaVac in PLWH

We further evaluated the immunogenicity of CoronaVac for different immune statuses. The definition of an immunological responder has been a confounding matter, in that different criteria are used by different researchers. In this study, PLWH with CD4^+^ T cell counts ≥ 350 cells/µL were defined as immunological responders. The results showed that the S-RBD-IgG titers of immunological responders (CD4^+^ T cell counts ≥ 350 cells/µL) was significantly higher than that of poor immunological responders (CD4^+^ T cell counts < 350 cells/µL) (22.4 U/mL (IQR, 17–24.4) vs. 11.2 U/mL (IQR, 4.6–21.2), *p* = 0.023, Figure 2F,G), and two groups were well matched in age and time since whole-course vaccination (*p* = 0.346 and *p* = 0.235, respectively). Thus, the CoronaVac vaccine was more likely to elicit lower humoral immune responses in poor immunological responders.

## 4. Discussion

This study was conducted when the coronavirus outbreak in China was largely under control and the free vaccination policy was implemented. The results indicate that PLWH have more vaccine hesitancy. COVID-19 vaccine hesitancy was driven primarily by safety and efficacy concerns. The results of adverse effects revealed that COVID-19 vaccines led to a tolerable safety profile in PLWH. Our data also showed that PLWH have a comparable immune response following CoronaVac vaccinations compared with HC, but poor immunological response might be associated with impaired humoral response in PLWH.

In the general Chinese population, the hesitancy rate of COVID-19 vaccination was 17.75% under the free vaccination policy [10], which is lower compared with the vaccine hesitancy rate of PLWH in the present study (27.5%). The vaccination rate with a first dose in our study was 57.9%, which is significantly lower than that of adult residents reported by Beijing Daily in the same period (94.5%) [20]. This situation is also observed in other vaccine inoculations, such as vaccines for influenza, human papillomavirus, and hepatitis B virus [21,22,23]. In the previous studies, the rates of vaccine hesitancy among PLWH towards the COVID-19 vaccine ranged from 28.7% to 54% [12,24,25,26]. Individually, vaccine hesitancy rates in PLWH were highest in black Americans (54%) [25] and were lowest in the French PLWH (28.7%) [12]. Nevertheless, we should be cautious when comparing vaccine hesitancy rates across regions because the influence of the vaccine type available in a study setting and different definition of vaccine hesitancy should not be overlooked.

We conducted univariate analyses for factors associated with vaccine hesitancy in PLWH. The demographic characteristics, HIV characteristics, and self-rated health status were not significantly associated with vaccine hesitancy. Of importance, PLWH with vaccine hesitancy were less likely to consult physicians than those without vaccine hesitancy. Evidence suggests that patients whose physicians recommend a vaccine are more likely to become vaccinated than patients who do not [27]. Most patients actively seek information about the vaccine and value their physician’s opinion in this area. This finding has been also confirmed by our study, which underlines the role of physicians in encouraging vaccine acceptance among patients.

Next, the perceived barriers against COVID-19 vaccination found in this study, namely concerns about safety and efficacy, have likewise been reported in other studies related to the introduction of a COVID-19 vaccine [12,28]. Feng et al. evaluated the safety of BBIBP-CorV inactivated vaccine in Chinese PLWH who are stable on ART with CD4^+^ T cell counts >200 cells/µL and their results were satisfactory [29]. The ChAdOx1 nCoV-19 (AZD1222) vaccine (an adenovirus-vectored vaccine) and the BNT162b2 mRNA vaccine also showed favorable safety among PLWH in South Africa and America, respectively [30,31]. To provide more evidence on the safety of inactivated COVID-19 vaccines, we evaluated the adverse reaction rates of two inactivated COVID-19 vaccines in PLWH that had favorable safety profiles in the general population [13,14]. Similarly, our data suggested that the adverse reactions were mild and self-limiting. No unexpected safety issues were found, and the adverse reaction profile observed was consistent with that previously reported for inactivated vaccines and other kinds of COVID-19 vaccines, such as the BNT162b2 mRNA COVID-19 vaccine [32]. All of the above results suggest that the safety of these two kinds of inactivated COVID-19 vaccines in PLWH is tolerable.

Moreover, as specific indexes for the evaluating effect of ART, the impact of a COVID-19 vaccine on CD4^+^ T cell counts and VL is not conclusive. We further measured the changes in CD4^+^ T cell counts and VL before and after vaccination, and the results demonstrated that a COVID-19 vaccine had no negative impact on either CD4^+^ T cell counts or VL during the study period. Furthermore, the CD4^+^ T cell counts of vaccinated PLWH were significantly increased. Based on previous studies on other vaccines, we speculate that the proliferation of CD4^+^ T cells may be relevant to the generation of a virus-specific neutralizing antibody [33]. However, the exact underlying mechanism needs to be further investigated.

We next examined whether the CoronaVac vaccine can elicit a similar humoral response in PLWH compared with HC. Our results supported recent reports that humoral responses elicited by COVID-19 vaccines are comparable in PLWH and HC within 4 weeks [23,24,25], and we further demonstrated a similar outcome over a longer period of time, supporting the current advice for PLWH to be immunized with COVID-19 vaccines. In previous studies, PLWH with CD4^+^ T cells counts <200 cells/µL have shown diminished SARS-CoV-2 antibody production after acute infection [34], as well as blunted immune responses to multiple vaccine types [35]. Our data showed that poor immunological response was associated with significantly lower S-RBD-IgG levels, suggesting that the impaired humoral response of COVID-19 vaccine in PLWH is possibly related to CD4^+^ T cell counts. In line with previous studies, CD4^+^ T cells, especially T-follicular helper (Tfh) cells, are required for the induction of high-affinity antibody responses and the formation of long-lived B cell memory. The structural changes in the germinal center and functionally altered Tfh cells derived by HIV replication and the consequent impaired interaction between Tfh cells and germinal center B cells might contribute to impaired immune response [36,37]. In addition, a third booster shot of a COVID-19 vaccine was reported to potentially provide more protection in the general population [38]. Whether adding additional doses to poor immunological responders is worthwhile needs to be further investigated.

Our study has several limitations. First, this study was an observational study over a short period, and the sample size was small. Second, the single-center on-spot survey resulted in sampling bias, for example, most participants were male and highly educated, and had good adherence to ART, so the results might not be generalizable to a random population sample. Third, limited by the natural characteristics of a cross-sectional study, the data on adverse reactions after vaccination were collected through the patients’ memory, which might cause ambiguous information and need to be verified in large prospective studies.

## 5. Conclusions

In summary, we found that the rate of COVID-19 vaccine hesitancy in adult PLWH on ART with virological suppression was lower than that in the general population. Evidence of the safety and efficacy of COVID-19 vaccines are key to enhancing the rates of vaccine coverage. Overall, an inactivated COVID-19 vaccine is safe and tolerable. It is not associated with HIV RNA rebound but might increase CD4^+^ T cells counts. Finally, our results add to a growing body of evidence that PLWH develop similar humoral immune responses to an inactivated COVID-19 vaccine compared with the general population, but poor immunological responders might need more effective vaccination strategies.

## Figures and Tables

**Figure 1 vaccines-09-01458-f001:**
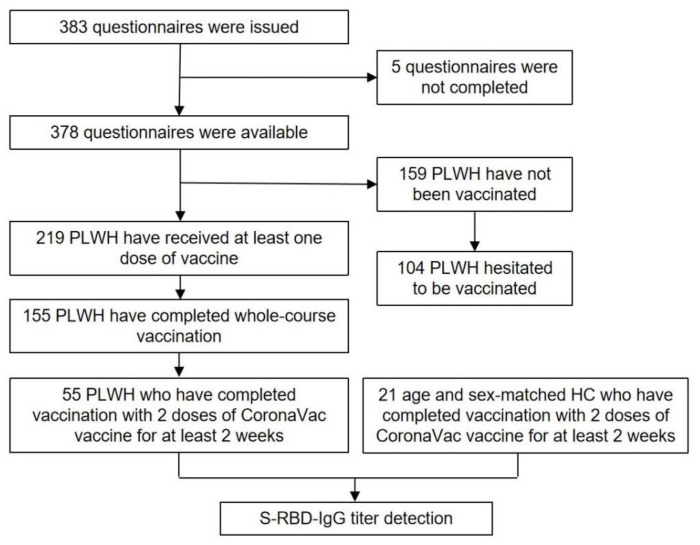
Flowchart of participants inclusion. PLWH: people living with HIV; HC: health control; S-RBD-IgG: spike receptor binding domain-protein specific IgG.

**Figure 2 vaccines-09-01458-f002:**
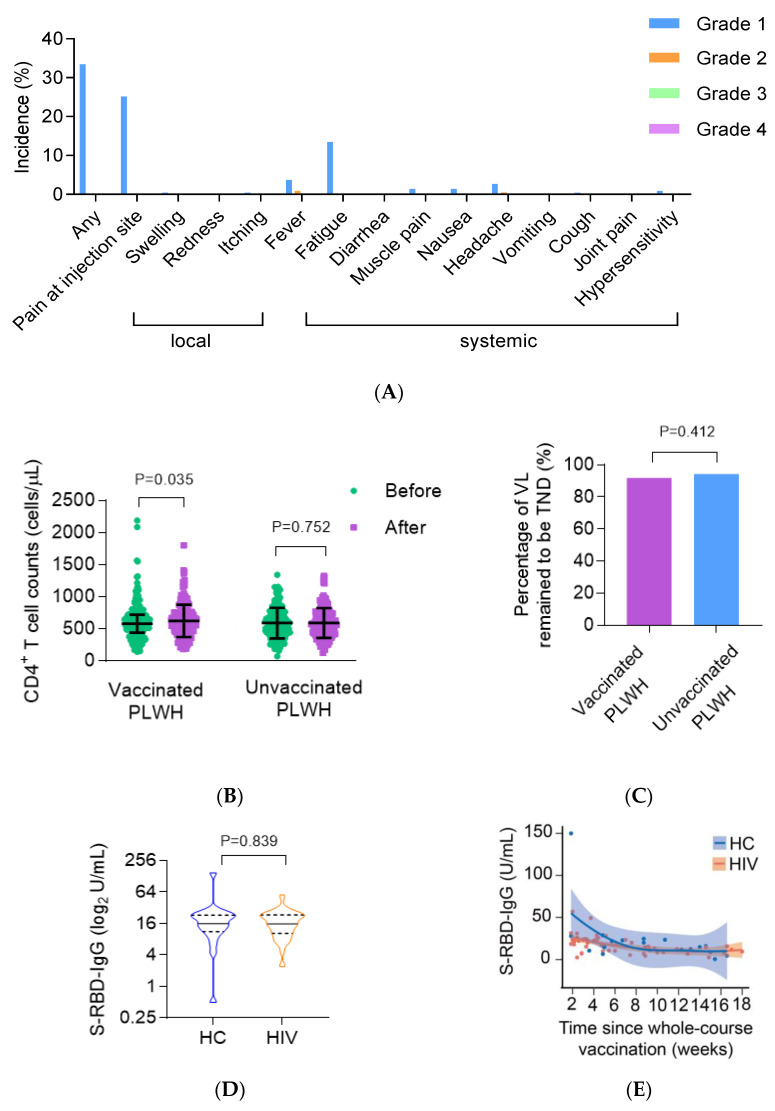
The safety and immunogenicity of COVID-19 vaccine and impact of COVID-19 vaccine on ART efficacy in PLWH. (**A**) Histogram demonstrating the incidence of local and systemic adverse reactions reported within 28 days after each dose of COVID-19 vaccine in PLWH (*n* = 219). (**B**) CD4^+^ T cell counts measured before and after vaccination against COVID-19 or during the same period in vaccinated PLWH (*n* = 219) and unvaccinated PLWH (*n* = 159). (**C**) The percentage of VL remained to be “TND” in vaccinated PLWH (*n* = 219) and unvaccinated PLWH (*n* = 159). (**D**) S-RBD-IgG titers after the second dose of the CoronaVac vaccine in PLWH group (*n* = 55) and HC group (*n* = 21). (**E**) Line plot demonstrating the dynamic trend of S-RBD-IgG titers (loess smoothed normalized counts SE) over time after the second dose of the CoronaVac vaccine in PLWH group (*n* = 55) and HC group (*n* = 21). (**F**) Line plot demonstrating the dynamic trend of S-RBD-IgG titers (loess smoothed normalized counts SE) over time after the second dose of the CoronaVac vaccine in immunological responders (CD4^+^ T cell counts > 350 cells/μL, *n* = 31) and poor immunological responders (CD4^+^ T cell counts ≤ 350 cells/μL, *n* = 8). (**G**) S-RBD-IgG titers after the second dose of the CoronaVac vaccine in immunological responders (CD4^+^ T cell counts >350 cells/μL, *n* = 31) and poor immunological responders (CD4^+^ T cell counts ≤ 350 cells/μL, *n* = 8). PLWH: people living with HIV; HC: health control; S-RBD-IgG: spike receptor binding domain-protein specific IgG.

**Table 1 vaccines-09-01458-t001:** Demographics, HIV characteristics, and health status of PLWH (*n* = 378).

Variables	Statistic Value
Sex (*n*, %)	
Male	374 (98.9%)
Female	4 (1.1%)
Age (years, *n*, %)	
18–30	100 (26.7%)
31–40	192 (51.3%)
41–50	56 (15.0%)
51–60	26 (7.0%)
Marital status (*n*, %)	
Unmarried	279 (76.2%)
Married	69 (18.8%)
Divorced/widowed	18 (4.9%)
Educational background (*n*, %)	
High school and below	70 (19.1%)
College/undergraduate	246 (67.2%)
Postgraduate and above	50 (13.7%)
Occupation (*n*, %)	
Business/service staff	129 (35.1%)
Professional and technical personnel	76 (20.7%)
Public official	36 (9.8%)
Farmer/industrial worker	21 (5.7%)
Student	6 (1.6%)
Unemployed	12 (3.2%)
Others	88 (23.9%)
Duration of ART treatment (years, median and IQR)	4.3 (2.8–6.0)
ART regimen (*n*, %)	
EFV + TDF + 3TC	378 (100%)
CD4^+^ T cell counts prior to ART initiation (cells/L, median and IQR)	305 (203–433)
VL prior to ART initiation (log_10_ copies/mL, median and IQR)	4.67 (4.16–5.01)
CD4^+^ T cell counts before 6 months (cells/L, *n*, %)	
<200	10 (2.6%)
200–349	52 (13.8%)
350–500	70 (18.5%)
>500	246 (65.1%)
Mode of HIV transmission (*n*, %)	
MSM	275 (72.7%)
Heterosexual sex	33 (8.7%)
Transfusion	9 (2.3%)
Others/unknown	61 (16.1%)
Scores of SF-12 (median and IQR)	
PCS	53 (47–55)
MCS	53 (46–56)

IQR: interquartile intervals; ART: antiviral therapy; EFV: efavirenz; TDF: tenofovir disoproxil fumarate; 3TC: lamivudine; VL: HIV RNA viral load; MSM: men who have sex with men; SF-12: 12-item short form health survey; PCS: physical component summary; MCS: mental component summary.

**Table 2 vaccines-09-01458-t002:** COVID-19 vaccination statuses and attitudes toward the COVID-19 vaccine in the vaccinated PLWH (*n* = 219).

**Patients Received at Least One Dose**	***n* (%)**
Manufacturers	
CoronaVac vaccine (Sinovac Life Sciences, Beijing, China)	128 (58.4%)
BBIBP-CorV vaccine (Beijing Institute of Biological Products, Beijing, China)	87 (39.7%)
Recombinant protein subunit vaccine (Anhui Zhifei Longcom Biopharmaceutical, Anhui, China)	3 (1.4%)
Recombinant adenovirus type-5 vectored vaccine (CanSino Biologics, Tianjin, China)	1 (0.5%)
Vaccinees who do not have concerns about vaccine	202 (92.2%)
Vaccinees who completed whole-course vaccination	155 (70.7%)

**Table 3 vaccines-09-01458-t003:** The reasons for not becoming vaccinated in the unvaccinated PLWH (*n* = 159).

**The Reasons for Not Having Been Vaccinated**	** *n* ** **(%)**
Concerns about side effects and/or poor efficacy	89 (56.0%)
Waiting to be scheduled.	31 (19.5%)
Contraindications for the vaccine	21 (13.8%)
No perceived need for vaccination	15 (9.4%)
Scheduling conflicts	3 (1.9%)

**Table 4 vaccines-09-01458-t004:** Probable effect factors associated with vaccine hesitancy of PLWH.

	Vaccine Acceptance (*n* = 274)	Vaccine Hesitancy (*n* = 104)	*p* Value
Age (*n*, %)			0.625
21–30	25 (24.3%)	75 (27.7%)	
31–40	5 (54.4%)	136 (50.2%)	
41–50	13 (12.6%)	43 (15.9%)	
51–60	9 (8.7%)	17 (6.3%)	
Marital status (*n*, %)			0.560
Unmarried	78 (77.2%)	201 (75.8%)	
Married	20 (19.8%)	49 (18.5%)	
Divorced/widowed	3 (3.0%)	15 (5.7%)	
Educational background (*n*, %)			0.648
High school and below	21 (20.6%)	201 (75.8%)	
College/undergraduate	68 (66.7%)	49 (18.5%)	
Postgraduate and above	13 (12.7%)	15 (5.7%)	
Occupation (*n*, %)			0.621
Business/service staff	29 (28.4%)	100 (37.6%)	
Professional and technical personnel	24 (23.5%)	52 (19.5%)	
Public officials	9 (8.8%)	27 (10.2%)	
Farmer/worker	7 (6.9%)	14 (5.3%)	
Students	1 (1.0%)	5 (1.9%)	
Unemployed	3 (2.9%)	9 (3.4%)	
Others	29 (28.4%)	59 (22.2%)	
Duration of ART treatment (years, median and IQR)	4.6 (2.9–5.9)	4.1 (2.8–6.1)	0.696
CD4^+^ T cell counts prior to ART initiation (cells/μL, median and IQR)	305 (183–454)	306 (213.5–425.5)	0.567
VL load prior to ART initiation (log_10_ copies/mL, median and IQR)	4.72 (4.39–5.16)	4.65 (3.95–4.97)	0.103
CD4^+^ T cell counts before 6 months (*n*, %)			0.505
<200	2 (1.9%)	2 (0.7%)	
200–349	16 (15.4%)	32 (11.7%)	
350–500	23 (22.1%)	57 (20.8%)	
>500	63 (60.6%)	183 (66.8%)	
Mode of HIV transmission (*n*, %)			0.094
MSM	72 (69.2%)	203 (74.1%)	
Heterosexual sex	7 (6.7%)	26 (9.5%)	
Transfusion	1 (1%)	8 (2.9%)	
Others/unknown	24 (23.1%)	37 (13.5%)	
Scores of SF-12 (median and IQR)			
PCS	53 (48.5–55)	54 (47–55)	0.159
MCS	53 (47.5–56)	53 (45–56)	0.593
Consulted physicians (*n*, %)	38 (36.5%)	148 (54%)	0.002

IQR: interquartile intervals; ART: antiviral therapy; MSM: men who have sex with men; SF-12: 12-item short form health survey; PCS: physical component summary; MCS: mental component summary.

**Table 5 vaccines-09-01458-t005:** Characteristics of PLWH and HC who had vaccinated with two doses of CoronaVac vaccine for at least 2 weeks.

	HC (*n* = 21)	PLWH (*n* = 55)	*p* Value
Age (years, mean ± SD)	35 ± 8	36 ± 11	0.681
Male (*n*, %)	17 (100%)	55 (100%)	1.000
Time since whole-course vaccination (weeks, median and IQR)	4.86 (3–9.14)	5 (3.71–8.71)	0.921
Vaccination interval (weeks, mean ± SD)	3.67 ± 0.8	3.64 ± 0.84	0.969
CD4^+^ T cell counts after vaccination (cells/L, mean ± SD)	769 ± 262	572 ± 203	0.001

## Data Availability

Data was obtained from Beijing Ditan Hospital, Capital Medical University and are available from the corresponding author with the permission of Beijing Ditan Hospital, Capital Medical University.

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
