# Peer review of "COVID-19 Vaccination in People Living with HIV (PLWH) in China: A Cross Sectional Study of Vaccine Hesitancy, Safety, and Immunogenicity"

_vaccines, 2021, doi:10.3390/vaccines9121458_

Round 1

Reviewer 1 Report

Liu et al. conducted a questionnaire-based survey to assess the efficacy and safety of two COVID-19 inactivated vaccines (used mainly in China) among a small group of people living with HIV (PLWH). They also fruitfully explored vaccine hesitancy in this vulnerable population group. The paper is quite well structured and clear; the statistical methodology is simple but sound. Authors report that PLWH has comparable safety and immune response after inactivated vaccination than healthy controls, even if a lower CD4+ T cells count was associated with lower immunological responses.

Some revisions are requested in order to improve the study: details are reported below as general and specific comments.

  • Please, add details on the study design in the title and the first paragraph of the methods section.
  • Please, specify relevant dates in the methods section: periods of recruitment and data collection. Also, add details on the recruitment setting of the included outpatients.
  • In the methods section, please explain how missing data were addressed in the analysis.
  • Please, specify if the inclusion criterion “have no epidemiologic history of COVID-19” includes having no confirmed contacts with infected individuals.
  • I believe authors should add references to the validated scales used in the questionnaire (SF-12, PCS and MCS).
  • I believe authors could expand a bit more both in the discussion section the possible adverse events to COVID-19 vaccination in observational studies compared with other similar scenarios (please, consider citing Vigezzi GP, Lume A, Minerva M, Nizzero P, Biancardi A, Gianfredi V, Odone A, Signorelli C, Moro M. Safety surveillance after BNT162b2 mRNA COVID-19 vaccination: results from a cross-sectional survey among the staff of a large Italian teaching hospital. Acta Biomed [Internet]. 2021 Oct. 1 [cited 2021 Nov. 23];92(S6):e2021450.).
  • Please consider making explicit the abbreviation OD at line 119.
  • I noted some discrepancies in the citation style and the references. Please emend the issues and update the bibliography (example in line 207).
  • Please check the content of 200-201 lines and the caption of Figure 2.
  • Authors might consider adding the same specific limitation of their work, such as the small sample size, which mainly included males and only four females.
  • Finally, even if the response rate authors obtained is exceptionally high, a fair generalisability of the obtained results must be cautious as the inactivated COVID-19 vaccines used have not been still authorized in Western countries, posing issues in the implementation of worldwide vaccination campaigns. I believe authors should stress more the importance of the vaccine type available in the study setting as the vaccine type (e.g., inactivated) could contribute to the observed hesitancy among PLWH, making comparisons with the acceptance rate in the same group in other countries to tackle this crucial element.
  • English revision is needed.

Author Response

Point 1: Please, add details on the study design in the title and the first paragraph of the methods section.

Response 1: We are grateful for the suggestion. We modified the title to “COVID-19 vaccination in people living with HIV (PLWH) in China: a cross sectional study of vaccine hesitancy, safety and immunogenicity”, and revised the first paragraph of the methods in the revised manuscript (L72).

Point 2: Please, specify relevant dates in the methods section: periods of recruitment and data collection. Also, add details on the recruitment setting of the included outpatients.

Response 2: 1) We added the information in the revised manuscript. The participants were recruited from 20 July to 4 August, 2021 and the data were collected from 20 July to 20 August, 2021 (L91). 2) We used two methods to recruit participants. We approached patients in the out-patient clinic in person and invited them to participate. If they agreed and were eligible, we provided a private room in which the participants and the research assistant could interact, and participants then completed a paper-based questionnaire. On the other hand, we recruited age- and sex-matched HC who had been vaccinated with two doses (0.5 ml/dose) of CoronaVac (Sinovac Life Sciences, Beijing, China) for at least 2 weeks by advertisements on the Internet. (L85). We have added details in the part of “methods” (L83-L91).

Point 3: In the methods section, please explain how missing data were addressed in the analysis.

Response 3: We gratefully appreciate for your valuable suggestion. According to the literature [1], cases with missing values were excluded from analysis if missing values did not exceed 5%. The maximum percentage of missing values was 3.2% (n = 12) in the present study and the missing value was excluded for analysis. We have revised in the part of “methods” (L140).

Reference

  • Rubin DB. Inference and missing data. Biometrika 1976, 63, :581–92.

Point 4: Please, specify if the inclusion criterion “have no epidemiologic history of COVID-19” includes having no confirmed contacts with infected individuals.

Response 4: According to the reviewer’s suggestion, PLWH were eligible if they met the following inclusion criteria: (1) 18-60 years old; (2) have been receiving a stable ART regimen for at least 1 year with an VL ≤50 copies/mL; (3) have no COVID-19 infection history and no contact history, including close or indirect contact with a person with a confirmed COVID-19 infection; (4) completed the questionnaire; and (5) signed written informed consent. We have added the information in methods section (L78-L82).

Point 5: I believe authors should add references to the validated scales used in the questionnaire (SF-12, PCS and MCS).

Response 5: We added proper reference about SF-12, PCS and MCS in the revised manuscript (L110).

Point 6: I believe authors could expand a bit more both in the discussion section the possible adverse events to COVID-19 vaccination in observational studies compared with other similar scenarios (please, consider citing Vigezzi GP, Lume A, Minerva M, Nizzero P, Biancardi A, Gianfredi V, Odone A, Signorelli C, Moro M. Safety surveillance after BNT162b2 mRNA COVID-19 vaccination: results from a cross-sectional survey among the staff of a large Italian teaching hospital. Acta Biomed [Internet]. 2021 Oct. 1 [cited 2021 Nov. 23];92(S6):e2021450.).

Response 6: According to the suggestion of review, we have further revised the discussion about the possible adverse events to COVID-19 vaccination. No unexpected safety issues were found, and the adverse reaction profile observed was consistent with that previously reported for inactivated vaccines and other kinds of COVID-19 vaccines, such as the BNT162b2 mRNA COVID-19 vaccine [2]. We have cited the reference in the part of “discussion” (L319).

Reference

[2] Vigezzi, G.P.; Lume, A.; Minerva, M.; Nizzero, P.; Biancardi, A.; Gianfredi, V.; Odone, A.; Signorelli, C.; Moro, M. Safety surveillance after BNT162b2 mRNA COVID-19 vaccination: results from a cross-sectional survey among staff of a large Italian teaching hospital. Acta Biomed 2021, 92, e2021450

Point 7: Please consider making explicit the abbreviation OD at line 119.

Response 7: Thank you so much for your careful check. We have added the abbreviation in revised manuscript (L136).

Point 8: I noted some discrepancies in the citation style and the references. Please emend the issues and update the bibliography (example in line 207).

Response 8: Thank you for pointing out this problem in manuscript. We have thoroughly checked the citation style and references.

Point 9: Please check the content of 200-201 lines and the caption of Figure 2.

Response 9: According to the reviewer’s suggestion, we further labeled the local and system adverse reaction in the Figure 2A and revised corresponding Figure legend (revised Fig. 2A and L258).

Point 10: Authors might consider adding the same specific limitation of their work, such as the small sample size, which mainly included males and only four females.

Response 10: Thank you for pointing out this problem in manuscript. We further discussed the limitation of the present study as follow: First, this study was an observational study over a short period, and the sample size was small. Second, the single-center on-spot survey resulted in sampling bias, for example, most participants were male and highly educated, and had good adherence to ART, so the results might not be generalizable to a random population sample. Third, limited by the natural characteristics of a cross-sectional study, the data on adverse reactions after vaccination were collected through the patients' memory, which might cause ambiguous information and need to be verified in large prospective studies. We have revised the manuscript in the part of “discussion” (L350-L357).

Point 11: Finally, even if the response rate authors obtained is exceptionally high, a fair generalisability of the obtained results must be cautious as the inactivated COVID-19 vaccines used have not been still authorized in Western countries, posing issues in the implementation of worldwide vaccination campaigns. I believe authors should stress more the importance of the vaccine type available in the study setting as the vaccine type (e.g., inactivated) could contribute to the observed hesitancy among PLWH, making comparisons with the acceptance rate in the same group in other countries to tackle this crucial element.

Response 11: Thank you for your advice. In the previous studies, the rates of vaccine hesitancy among PLWH towards the COVID-19 vaccine ranged from 28.7% to 54% [3-6]. Individually, vaccine hesitancy rates in PLWH were highest in black Americans (54%) [4], and were lowest in the French PLWH (28.7%) [5]. Nevertheless, we should be cautious when comparing vaccine hesitancy rates across regions because the influence of the vaccine type available in a study setting and different definition of vaccine hesitancy should not be overlooked. We have revised in discussion section (L290-L296).

References

[3] Ekstrand, M.L.; Heylen, E.; Gandhi, M.; Steward, W.T.; Pereira, M.; Srinivasan, K. COVID-19 Vaccine Hesitancy Among PLWH in South India: Implications for Vaccination Campaigns. J Acquir Immune Defic Syndr 2021, 88, 421-425.

[4] Bogart, L.M.; Ojikutu, B.O.; Tyagi, K.; Klein, D.J.; Mutchler, M.G.; Dong, L.; Lawrence, S.J.; Thomas, D.R.; Kellman, S. COVID-19 Related Medical Mistrust, Health Impacts, and Potential Vaccine Hesitancy Among Black Americans Living With HIV. J Acquir Immune Defic Syndr 2021, 86, 200-207.

[5] Vallee, A.; Fourn, E.; Majerholc, C.; Touche, P.; Zucman, D. COVID-19 Vaccine Hesitancy among French People Living with HIV. Vaccines (Basel) 2021, 9.

[6] Jones, D.L.; Salazar, A.S.; Rodriguez, V.J.; Balise, R.R.; Starita, C.U.; Morgan, K.; Raccamarich, P.D.; Montgomerie, E.; Nogueira, N.F.; Barreto, O.I.; et al. Severe Acute Respiratory Syndrome Coronavirus 2: Vaccine Hesitancy Among Underrepresented Racial and Ethnic Groups With HIV in Miami, Florida. Open Forum Infect Dis 2021, 8, b154.

Point 12: English revision is needed

Response 12: According to the reviewer’s suggestion, the manuscript has been polished by an English language editing company, so we hope it can meet the journal’s standard.

Reviewer 2 Report

Dear Editor,

I read with interest the manuscript by Ying Liu et al. entitled "COVID-19 vaccination in people living with HIV (PLWH): vaccine hesitancy, safety and immunogenity."
I think the topic is of great interest and current in the debate among the scientific community.
I consider it very well done and suitable for publication after making some minor corrections.

Firstly, in light of the methods and limitations well pointed out (L320), I believe it is important to include in both the title and the abstract indications about the specific population being studied. For example: COVID-19 vaccination in people living with HIV (PLWH) in China: vac- 2
cine hesitancy, safety and immunogenity.

Whenever reference is made to the General Population when citing studies, I recommend that the specific country of that sample is also indicated in the text. We know how context-specific the VH phenomenon is. For example:
L269 and L328.

I recommend a careful linguistic review. For example, in Figure 1: 104 PLWH hesitated to get vaccinated or to get the vaccine.

The caption in Fig. 2 needs revision.

Author Response

Response to Reviewer 2 Comments

Point 1: Firstly, in light of the methods and limitations well pointed out (L320), I believe it is important to include in both the title and the abstract indications about the specific population being studied. For example: COVID-19 vaccination in people living with HIV (PLWH) in China: vaccine hesitancy, safety and immunogenicity.

Response 1: We are grateful for the suggestion. According to the reviewer’s suggestion, we have modified the title to “COVID-19 vaccination in people living with HIV (PLWH) in China: a cross sectional study of vaccine hesitancy, safety and immunogenicity” and revised abstract (L21).

Point 2: Whenever reference is made to the General Population when citing studies, I recommend that the specific country of that sample is also indicated in the text. We know how context-specific the VH phenomenon is. For example: L269 and L328.

Response 2: 1) Thank you for your comments. In the previous manuscript (L269), the general Chinese population were recruited and we have amended the manuscript in the part of “discussion” (L284). 2) In the previous manuscript (L328), we further revised the manuscript as follow: Feng et al. evaluated the safety of BBIBP-CorV inactivated vaccine in Chinese PLWH who are stable on ART with CD4+ T cell counts >200 cells/mL and their results were satisfactory [7]. The ChAdOx1 nCoV-19 (AZD1222) vaccine (an adenovirus-vectored vaccine) and the BNT162b2 mRNA vaccine also showed favorable safety among PLWH in South Africa and America, respectively [8,9]. We revised the manuscript in the part of “discussion” (L308-L313).

References:

[7] Feng Y, Zhang Y, He Z, Huang H, Tian X, Wang G, Chen D, Ren Y, Jia L, Wang W, et al. Immunogenicity and safety of an inactivated SARS-CoV-2 vaccine in people living with HIV-1 (preprint). Available online: https://doi.org/10.1101/2021.09.14.21263556 (assecced on 22 September 2021)., doi: 10.1101/2021.09.14.21263556.

[8] Madhi, S.A.; Koen, A.L.; Izu, A.; Fairlie, L.; Cutland, C.L.; Baillie, V.; Padayachee, S.D.; Dheda, K.; Barnabas, S.L.; Bhorat, Q.E.; et al. Safety and immunogenicity of the ChAdOx1 nCoV-19 (AZD1222) vaccine against SARS-CoV-2 in people living with and without HIV in South Africa: an interim analysis of a randomised, double-blind, placebo-controlled, phase 1B/2A trial. The Lancet HIV 2021, 8, e568-e580, doi: 10.1016/S2352-3018(21)00157-0.

[9] Woldemeskel, B.A.; Karaba, A.H.; Garliss, C.C.; Beck, E.J.; Wang, K.H.; Laeyendecker, O.; Cox, A.L.; Blankson, J.N. The BNT162b2 mRNA Vaccine Elicits Robust Humoral and Cellular Immune Responses in People Living With Human Immunodeficiency Virus (HIV). Clin. Infect. Dis. 2021, doi: 10.1093/cid/ciab648.

Point 3: I recommend a careful linguistic review. For example, in Figure 1: 104 PLWH hesitated to get vaccinated or to get the vaccine.

Response 3: Thank you so much for your careful check. We have revised “vaccined” to “vaccinated” in the revised Fig. 1 (Figure 1).

Point 4: The caption in Fig. 2 needs revision.

Response 4: We have deleted the instruction for figures and revised the caption of Figure 2 (L258).